# Mapping and characterising areas with high levels of HIV transmission in sub-Saharan Africa: A geospatial analysis of national survey data

Caroline A. Bulstra[1,2☯], Jan A. C. Hontelez[1,2☯]*, Federica Giardina[1], Richard Steen[1], Nico J. D. Nagelkerke[1], Till Bärnighausen[2,3,4], Sake J. de Vlas[1]

1 Department of Public Health, Erasmus MC, University Medical Center Rotterdam, Rotterdam, The Netherlands, 2 Heidelberg Institute of Global Health, Heidelberg University Medical Center, Heidelberg, Germany, 3 Department of Global Health and Population, Harvard T.H. Chan School of Public Health, Boston, Massachusetts, United States of America, 4 Africa Health Research Institute (AHRI), KwaZulu-Natal, South Africa

☯ These authors contributed equally to this work.
* j.hontelez@erasmusmc.nl

**Data Availability Statement:** All utilised data are open-source, and the hyperlinks to the different

## Abstract

### Background

In the generalised epidemics of sub-Saharan Africa (SSA), human immunodeficiency virus (HIV) prevalence shows patterns of clustered micro-epidemics. We mapped and characterised these high-prevalence areas for young adults (15–29 years of age), as a proxy for areas with high levels of transmission, for 7 countries in Eastern and Southern Africa: Kenya, Malawi, Mozambique, Tanzania, Uganda, Zambia, and Zimbabwe.

### Methods and findings

We used geolocated survey data from the most recent United States Agency for International Development (USAID) demographic and health surveys (DHSs) and AIDS indicator surveys (AISs) (collected between 2008–2009 and 2015–2016), which included about 113,000 adults—of which there were about 53,000 young adults (27,000 women, 28,000 men)—from over 3,500 sample locations. First, ordinary kriging was applied to predict HIV prevalence at unmeasured locations. Second, we explored to what extent behavioural, socioeconomic, and environmental factors explain HIV prevalence at the individual- and sample-location level, by developing a series of multilevel multivariable logistic regression models and geospatially visualising unexplained model heterogeneity. National-level HIV prevalence for young adults ranged from 2.2% in Tanzania to 7.7% in Mozambique. However, at the subnational level, we found areas with prevalence among young adults as high as 11% or 15% alternating with areas with prevalence between 0% and 2%, suggesting the existence of areas with high levels of transmission Overall, 15.6% of heterogeneity could be explained by an interplay of known behavioural, socioeconomic, and environmental factors. Maps of the interpolated random effect estimates show that environmental variables,

data sources are provided in the Methods section of the manuscript.

**Funding:** This study was funded by the Dutch AIDS Foundation (P-29702). Furthermore, JH was supported by the NWO Talent Scheme. Till Bärnighausen was supported by the Alexander von Humboldt Foundation through the Alexander von Humboldt Professor award, funded by the Federal Ministry of Education and Research; the Wellcome Trust; and from NICHD of NIH (R01-HD084233), NIA of NIH (P01-AG041710), NIAID of NIH (R01-AI124389 and R01-AI112339) as well as FIC of NIH (D43-TW009775). The funders of the study had no role in study design, data collection, data analysis, data interpretation, or writing of the report. The views expressed in this article are our own and not necessarily those of the funders.

**Competing interests:** The authors have declared that no competing interests exist.

**Abbreviations:** AIS, AIDS indicator survey; aOR, adjusted odds ratio; DHS, demographic and health survey; ELISA, enzyme-linked immunosorbent assay; EVI, enhanced vegetation index; GHF, global human footprint; HIV, human immunodeficiency virus; MOR, Median Odds Ratio; NASA, National Aeronautics and Space Administration; OR, odds ratio; RR, relative risk; SDG, Sustainable Development Goal; SSA, sub-Saharan Africa; STI, sexually transmitted infection; UNAIDS, Joint United Nations Programme on HIV/AIDS; USAID, United States Agency for International Development.

representing indicators of economic activity, were most powerful in explaining high-prevalence areas. Main study limitations were the inability to infer causality due to the cross-sectional nature of the surveys and the likely under-sampling of key populations in the surveys.

## Conclusions

We found that, among young adults, micro-epidemics of relatively high HIV prevalence alternate with areas of very low prevalence, clearly illustrating the existence of areas with high levels of transmission. These areas are partially characterised by high economic activity, relatively high socioeconomic status, and risky sexual behaviour. Localised HIV prevention interventions specifically tailored to the populations at risk will be essential to curb transmission. More fine-scale geospatial mapping of key populations,—such as sex workers and migrant populations—could help us further understand the drivers of these areas with high levels of transmission and help us determine how they fuel the generalised epidemics in SSA.

## Author summary

### Why was this study done?

- Previous studies showed that heterogeneity in human immunodeficiency virus (HIV) prevalence exists among the general population in Eastern and Southern Africa, the geographic area most severely affected by the HIV pandemic.

- Whereas HIV prevalence among adults does not reveal when persons have been infected, young adults are most likely recently infected, and therefore high-prevalence areas among this subpopulation can proxy locations of ongoing transmission.

- The location and underlying determinants of high HIV prevalence areas among young adults can help to shape spatially targeted and risk-group–tailored interventions to reduce transmission.

### What did the researchers do and find?

- We found clear areas of high prevalence in young adults in between vast regions with relatively low prevalence for all 7 countries in Eastern and Southern Africa.

- HIV prevalence in young adults was partly explained by an interplay of behavioural, socioeconomic, and environmental (i.e., economic activity) factors, and environmental factors were especially predictive of high transmission locations.

### What do these finding mean?

- Our findings, together with the existing evidence, indicate that key population dynamics, especially related to seasonal and economic migration and associated sex work, might play a major role in fuelling HIV transmission.

- In further reducing HIV transmission in Eastern and Southern Africa, areas of high HIV prevalence in young adults should be priority areas for tailored HIV prevention interventions towards reaching the fast-track commitments to end the HIV epidemic by 2030.

## Introduction

Sustainable Development Goal (SDG) 3, "to ensure healthy lives and promote well-being for all at all ages" [1], together with the Joint United Nations Programme on HIV/AIDS (UNAIDS) fast-track strategy, explicitly call to end the pandemic by 2030 [2]. In 2017, about 37 million people were living with human immunodeficiency virus (HIV) worldwide, 70% of whom were residing in sub-Saharan Africa (SSA) [3]. The countries in Eastern and Southern Africa are especially severely affected by the pandemic, with general population prevalences ranging from 5% in Tanzania to 27% in eSwatini (former Swaziland) [3]. Mounting evidence suggests that these HIV epidemics are heterogeneous [4,5] and that the transmission of HIV is largely concentrated across clustered micro-epidemics of different scales [6,7]. As these high-prevalence areas are likely important drivers of the epidemic [8,9], identifying their location and underlying determinants is essential to further optimise HIV prevention and treatment interventions.

Although mapping overall HIV prevalence in the adult population gives an adequate indication of treatment service needs [4,5], it is not straightforward to use such data to inform policy makers on areas with high levels of transmission, because many, especially older, adults were infected many years prior to any survey and possibly at other locations. Mapping heterogeneity of HIV prevalence in young adults, who are most likely to have been recently infected, will more directly pinpoint areas of high HIV transmission [10]. Furthermore, identifying underlying determinants of heterogeneity in HIV prevalence among young adults in the high endemic countries can help shape spatially targeted and risk-group–tailored interventions to reduce transmission.

We identified areas of high HIV prevalence in young adults (women 15–24 years and men 15–29 years) for 7 countries in Eastern and Southern Africa (Kenya, Malawi, Mozambique, Tanzania, Uganda, Zambia, and Zimbabwe), using geolocated HIV prevalence data from demographic and health surveys (DHSs) and AIDS indicator surveys (AISs). Next, we explored to what extent sexual behavioural, socioeconomic, and environmental factors explain the geospatial heterogeneity in young adults, as well as what heterogeneity remains unexplained.

## Methods

### Data

The Demographic and Health Survey Programme is a programme that conducts national, population-level DHSs worldwide, in which individuals are interviewed about a wide range of behavioural, socioeconomic, and epidemiological parameters. The AISs and many DHSs include voluntary HIV testing in adults. Approximately 350 sample locations (primary sampling units) are randomly sampled throughout the country of interest, and at each location residents of about 25 households are sampled. All individuals that were at home during one of the visits and were between 15 and 49 years of age (women) or 54 years of age (men) were eligible for the survey. GPS coordinates of sample locations are randomly displaced up to 2 km for

urban and up to 5 km for rural sample locations, to ensure confidentiality of participants. Sample weights are incorporated in the DHS to translate unbalanced sampling into national representative data. In our study, we used these sample weights to estimate national HIV prevalence among adults and young adults for all 7 countries. For the other analyses, we combined data of multiple countries and explored local-level variances and thus did not use the sampling weights. More details on survey protocols and questionnaires can be found on the DHS website (https://dhsprogram.com/).

We extracted data from countries in Eastern and Southern African that had a DHS or an AIS conducted and for which behavioural data, geographical coordinates of the sample locations, and HIV biomarker surveys were available. In case of multiple eligible surveys for a country, we selected the most recent one. The countries and surveys chosen for this study were Kenya (years 2008–2009), Malawi (2015–2016), Mozambique (2009), Tanzania (2011–2012), Uganda (2011), Zambia (2013–2014), and Zimbabwe (2015). The overall study area with DHS sample locations for each country included in the study are shown in S1 Fig. We selected all adults (women 15–49 years and men 15–54 years) and sub-selected young adults (women 15–24 years and men 15–29 years). The discrepancy in age cut-off reflects the common age difference in sexual debut and relationships between women and men [11].

A typical DHS data set contains over 250 variables. For the purpose of our study, we only extracted individual-level candidate variables that were deemed of interest, based on findings from previous studies [4,6,8,12–15]: age, sex, HIV status, educational level, wealth index, primary occupation, whether the person is a de jure household member (usual resident) or de facto household member (slept in the household last night) as a proxy for mobility, number of lifetime sexual partners, number of sexual partners during the past 12 months, having had a sexually transmitted infection (STI) or signs of an STI during the past 12 months, condom use during last intercourse, male circumcision, and being paid for sexual intercourse during the past 12 months (only men). HIV status is determined by testing a blood sample from a finger prick with an enzyme-linked immunosorbent assay (ELISA). The other variables were self-reported by participants via the survey questionnaires.

We used the following environmental variables at sample locations: urban versus rural classification of the cluster, population density, proximity to highways, proximity to cities with more than 250,000 inhabitants, proximity to border crossings and major ports, enhanced vegetation index (EVI), and global human footprint (GHF). EVI reflects the vegetation in an area, in which low values represent areas with no or little green vegetation (e.g., urban areas) and high values represent areas where vegetation is more abundant (e.g., agricultural land, forests, grassland), and can be used as a proxy for degree of urbanization [4]. GHF represents the relative human influence and economic activity, incorporating 9 layers covering infrastructure, land use, population density, and access (coastlines, roads, railroads, navigable rivers). Areas of large human influence and high levels of economic activity are characterised by high GHF values. Population density estimates were originally obtained from WorldPop (https://www.worldpop.org/), and data from 2010—the most recent year—were utilised. EVI and GHF were available from the National Aeronautics and Space Administration (NASA) Earth Observatory Group (http://sedac.ciesin.columbia.edu/). EVI data were available for 2010, and GHF data were available for 2005. Because DHS sample locations are, to some extent, randomly displaced, population density, EVI, and GHF at each un-displaced DHS sample location are also made available for each survey. Locations of major cities were extracted from the World Population Review website (www.worldpopulationreview.com/worldcities/), and highways were derived from GADM national infrastructure shape files (http://www.gadm.org/) based on Google Maps. Locations of border crossings and major ports were obtained through the Southern Africa Integrated Regional Transport Program report (2010) [16]. The shortest

Euclidean distances from each DHS sample location to the nearest highway, major city, and border crossing or port were calculated. An overview and description of all included variables can be found in S1 Table. Maps of the included environmental variables are provided in S2 Fig to S8 Fig.

## Statistical analyses

Our study was explorative in nature, and we did not have a formal prespecified analysis plan. First, we determined the spatial distribution in HIV prevalence among adults and young adults by interpolating logit-transformed DHS sample location-level HIV prevalence data using ordinary kriging [17]. Kriging is an interpolation method based on the spatial autocorrelation of variables [18]. Spatial autocorrelation was measured by means of Moran's $I$, using the inverse distance between sample locations as weights. Spatial autocorrelation structures were obtained through semivariogram modelling, in which the average squared difference in HIV prevalence between each pair of data points (on the y-axis) is plotted against the corresponding distance between the point-pairs (on the x-axis). The overall relation between HIV prevalence and distance was estimated by fitting an exponential curve through these points. We used this to create continuous surface maps of HIV prevalence, in which the HIV prevalence at each 5 $\text{km}^2$ grid cell was estimated using the aforementioned method. The equations and model estimates are provided in S1 Equations. To enhance the power of our study, we decided to not stratify the kriging by sex in the main analysis. However, we also present sex-specific maps of kriged HIV prevalence (S9 Fig). We compared the sex-specific surfaces of HIV prevalence by means of mapping the square root of the squared difference in HIV prevalence (per 5 $\text{km}^2$ grid cell), to illustrate the absolute differences in HIV prevalence (S10 Fig panel A), and by plotting the predicted HIV prevalence (per 5 $\text{km}^2$ grid cell) of women against the predicted HIV prevalence of men (S10 Fig panel C). Both comparisons show that there are only minor differences in terms of the locations of high HIV prevalence for both sexes.

Second, we developed a series of multiple multilevel logistic regression models to determine to what extent behavioural, socioeconomic, and environmental determinants of HIV can explain individual- and location-level HIV prevalence among young adults. Missing values (up to 2.6%) were checked to be missing at random and, if so, imputed using multiple imputation. First, bivariate associations were tested, and all variables with a $p$-value larger than 0.1 were excluded. We then developed multiple models using a stepwise approach. In the first step, we made an 'empty' model—with HIV prevalence as the dependent variable, with only age- and sex-fixed effects, and with location random effects as predictors. In the second step, we used stepwise forward selection to construct 3 separate multiple regression models for behavioural, socioeconomic, and environmental factors, respectively, out of the nested model. Likelihood tests were used to determine whether the addition of a variable improved the statistical fit of the regression models significantly ($p < 0.05$). In the third step, we used the same stepwise forward selection to construct a full model containing both behavioural, socioeconomic, and environmental factors. We did not adjust for country-level confounding in the main analysis because we expect the associations between HIV and the predictor variables to be similar across countries. However, we did perform a sensitivity analysis in which we added country-fixed effects to the final model. We compared the marginal and conditional $R^2$ of the models at each step. The marginal $R^2$ indicates how much of the HIV heterogeneity is explained by the fixed factors in the model. The conditional $R^2$ represents the amount of heterogeneity explained by both fixed and random factors in the model. By comparing both $R^2$ values, we assessed how much of HIV heterogeneity is explained by the fixed factors in the models and how much is additionally captured by the location-level random effect in each model [19]. We

translated variance of random effects into Median Odds Ratios (MORs) as an indicator of geographical heterogeneity. For each model, the MOR would be equal to 1.0 if there were no differences in probability of being HIV infected per sample location, and it can be interpreted as the increase in (median) HIV risk that is associated with moving from a location with a low random effect to a location with a high random effect. See elsewhere for a more detailed explanation [20]. The MOR equation is provided in S1 Equations. The final model is also fitted as modified Poisson (with robust variance) for easier interpretation of the estimates, here as relative risks (RRs) instead of odds ratios (ORs).

Third, we extracted and compared the location-specific random effect estimates from the empty model, the 3 separate models for behavioural, socioeconomic, and environmental factors, as well as the 'full' regression model. We kriged random effect estimates from the 5 models to visualise to what extent the variables in the different models explain the geospatial HIV heterogeneity and areas with a high HIV prevalence.

Next to the main analyses, we also performed an internal validation of our regression models and an external validation of our kriging results. The internal validation was done using nonrandom cross-validation [21,22] to check, for each country individually, how much heterogeneity (indicated by the conditional $R^2$) was explained by the final regression model. For the external validation, we searched peer-reviewed literature reporting on age-stratified HIV prevalence estimates in population-based cohorts, such as those part of the ALPHA network [23], situated in one of the countries of our study. We compared our predicted HIV prevalence among young adults, obtained through kriging, to the estimates from these cohorts.

All analyses were done using ArcGIS Pro version 2.3 and R version 3.4.3. Reporting of study design and analysis followed RECORD guidelines (S1 Checklist) [24].

### Ethical approval

All the utilised DHS and AIS data sets are publicly available, and the Demographic and Health Survey Programme de-identifies all data before making them available to the public. The geospatial data (WorldPop, NASA) do not contain variables at the level of human subjects. Therefore, this work did not require ethical approval.

### Results

Our study included 112,785 adults, of which there were 53,234 young adults (25,536 women, 27,698 men) from 3,665 different sample locations throughout the 7 countries. The number of individuals included in the study as well as location-level HIV prevalence in the study population for adults and young adults are provided in Fig 1. Among adults, the mean country-level HIV prevalence ranges from 5.4% (Tanzania) to 14.4% (Zimbabwe). Among young adults, mean country-level HIV prevalences are generally lower, ranging from 2.2% (Tanzania) to 7.7% (Mozambique), while the median HIV prevalences are (close to) zero. This reflects the fact that most sample locations have very low HIV prevalence, and only a minority of sample locations have high prevalence. HIV prevalence among adults and young adults is strongly spatially clustered ($p < 0.001$); the observed Moran's $I$ index values were 0.13 and 0.05, respectively (on a scale from −1, fully scattered, to 1, fully clustered). A detailed overview of the Moran's $I$ and (logit-transformed) HIV prevalence density plots can be found in S2 Table and S11 Fig and S12 Fig, respectively.

Fig 2 shows the geospatial distribution of HIV prevalence in adults (panel A) and young adults (panel B) in 7 countries of Eastern and Southern Africa. All countries showed substantial levels of heterogeneity in HIV prevalence at the subnational level. Overall, HIV prevalence is higher among adults—with high-prevalence areas in the same locations—but prevalence

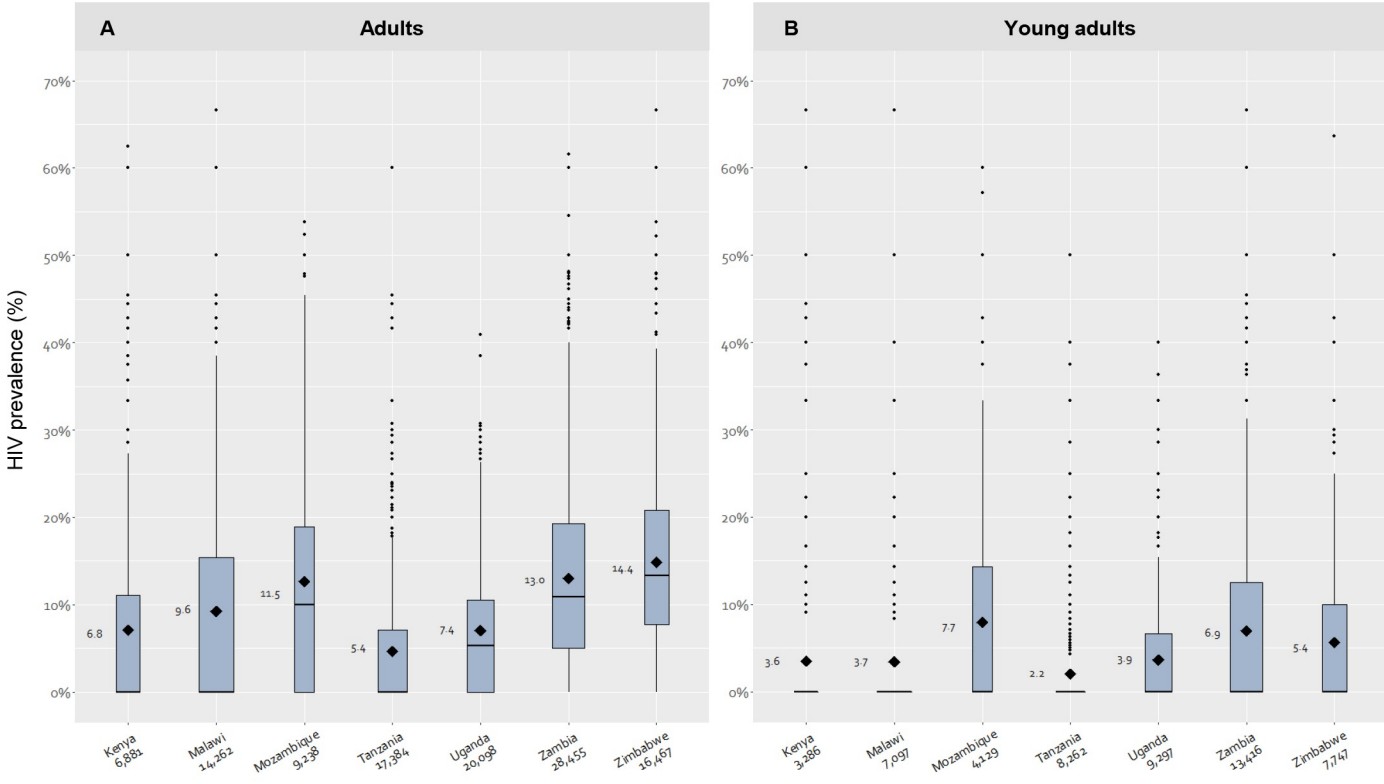

**Fig 1. Median, mean, and DHS sample location variance in national HIV prevalence estimates among adults (A) and young adults (B) for each country included in this study.** The mean, presented in the labels left of the bars, represents the weighted HIV prevalence per country. Data obtained through https://dhsprogram.com/. DHS, demographic and health survey; HIV, human immunodeficiency virus.

among adults is larger and more spread out compared to among young adults (illustrated by the large red and purple areas). Geospatial heterogeneity in HIV prevalence among young adults was more profound than among adults in most of the countries in our analysis, illustrated by clear concentrated micro-epidemics (red and purple areas) located in between areas of very low prevalence (white, yellow, and orange areas). In both Zambia and Zimbabwe, high-prevalence areas, of over 15% HIV prevalence in young adults, were found. The national HIV prevalence among young adults in Malawi is about 3.6%, yet our analysis identified areas where HIV prevalence reached levels of up to 11%, in particular around the highways and major cities in the south. Similarly, in Kenya HIV prevalence in young adults is about 3.9% nationally, yet prevalence around Lake Victoria reaches levels of over 15%. Maps and scatterplots illustrating the more detailed differences in HIV prevalence (per 5 km$^2$ grid cell) between adults and young adults are provided in S6 Fig.

According to the resulting best-fitting multiple multilevel logistic regression model on the association between HIV status and sexual behavioural variables in young adults, the following variables were strongly associated with being infected with HIV: 10 or more reported lifetime sex partners, an STI or STI symptoms over the past 12 months, condom use during last intercourse, and not being circumcised (men only). Educational level, wealth index, and occupation were variables associated with HIV in the final socioeconomic model. The highest level of education was most protective (adjusted odds ratio [aOR] 0.52 [0.26–0.78], $p < 0.001$), whereas being from asset quintile 4—the second wealthiest quintile—was associated with the highest

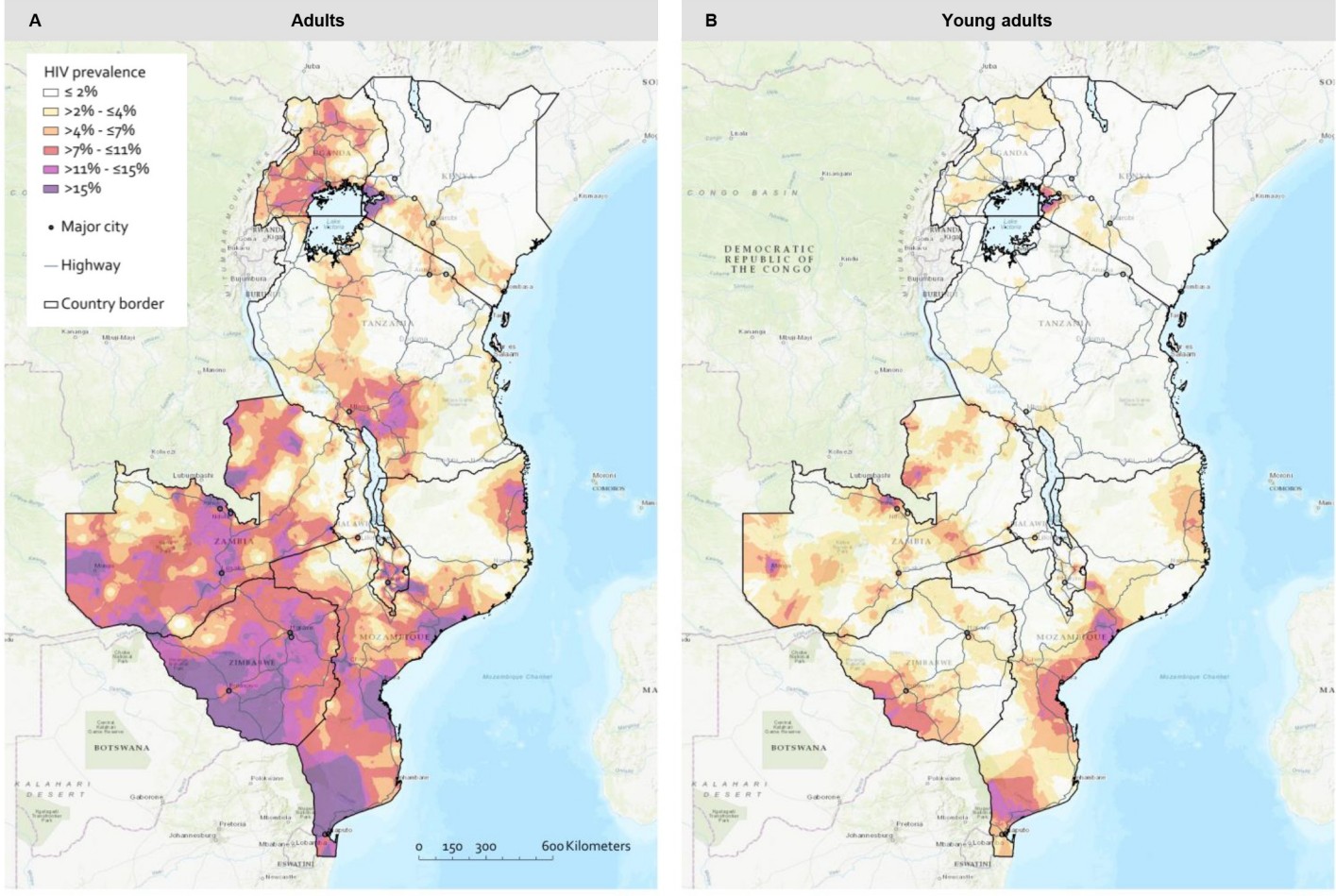

**Fig 2. Continuous surface maps of HIV prevalence in adults (women 15–49 years and men 15–54 years) (A) and young adults (women 15–24 years and men 15–29 years) (B) for 7 countries in Eastern and Southern Africa.** Predicted geographical distribution of HIV prevalence resulted from interpolating data on HIV prevalence in geolocated sample locations derived from the most recent DHS or AIS in each country, using ordinary kriging. Major cities (more than 250,000 inhabitants) are indicated on the maps. To enhance comparison between both panels, we applied the same legend for HIV prevalence levels. The HIV prevalence maps for women and men separately are provided in S4 Fig. Data obtained through https://dhsprogram.com/. AIS, AIDS indicator survey; DHS, demographic and health survey; HIV, human immunodeficiency virus.

risk of HIV (aOR 1.46 [1.31–1.62], $p < 0.001$). Several variables were significantly associated with HIV in the environmental model. Young adults living in rural sample locations were less likely to be infected with HIV than those with urban residence: in cities, the overall HIV prevalence among adults was over 7%, compared to below 4% in rural settings. Also, population density, proximity to nearest major city, EVI, and GHF at location of a DHS cluster showed a significant association with HIV. HIV prevalence levels were highest (about 6%) in areas with the highest population density (more than 500 people per $km^2$) but was also relatively high (about 5%) in areas with the lowest population density (less than 25 people per $km^2$). HIV prevalence levels did not differ considerably between sample locations with different levels of greenness (indicated by the EVI). Living in an area with a relatively high GHF, as a proxy for economic activity, was associated with a high HIV risk (aOR 1.68 [1.41–1.94], $p < 0.001$): HIV prevalence levels for the highest two levels of GHF were almost 7%, compared to around 4% at the lower levels. The best-fitting combined 'full' model contains both behavioural, socioeconomic, and environmental variables: lifetime number of sex partners, STIs, male circumcision,

**Table 1. Overview of the heterogeneity ($R^2$) explained by the best-fitting multilevel multiple logistic regression models and random-effects MOR.**

| | Conditional $R^2$: total heterogeneity explained by model (%) | Marginal $R^2$: heterogeneity explained by included fixed effects (%) | Random-effect $R^2$: location-level heterogeneity captured by model (%) | MOR |
|---|---|---|---|---|
| Nested 'empty' model | 26.3 | 7.2 | 19.1 | 2.41 |
| Sexual behavioural model | 27.8 | 10.2 | 17.6 | 2.35 |
| Socioeconomic model | 25.8 | 8.6 | 17.2 | 2.30 |
| Environmental model | 26.8 | 11.4 | 15.4 | 1.95 |
| Combined 'full' model | 29.6 | 15.6 | 14.0 | 1.94 |

**Abbreviation:** MOR, Median Odds Ratio

education, type of residence, EVI, and GHF. A complete overview of the bivariate models, nested model (only adjusted for age and sex), best-fitting models, and final model can be found in S3 to S8 Tables and S13 Fig. Finally, the combined 'full' model was also fitted as a modified Poisson regression model, resulting in adjusted relative risks that were comparable to the aORs from the logistic regression model (S9 and S10 Tables).

The results in Table 1 show that 7.2% (marginal $R^2$: e.g., percentage of heterogeneity explained by fixed effects in the model) of the 26.3% (conditional $R^2$: e.g., percentage of heterogeneity explained by both fixed and random effects in the model) of the HIV heterogeneity could be explained by age and sex alone ('nested' model). The remaining variance captured by the model (19.1%) is attributed to location-level random effects. Environmental fixed effects explain most of the heterogeneity in HIV prevalence among young adults (marginal $R^2$ 11.4%, conditional $R^2$ 26.8%), higher than behavioural (marginal $R^2$ 10.2%, conditional $R^2$ 27.8%) or socioeconomic (marginal $R^2$ 8.6%, conditional $R^2$ 25.8%) fixed effects. According to the $R^2$ of the combined 'full' best-fitting model, HIV heterogeneity could be best explained (marginal $R^2$ 15.6%, conditional $R^2$ 29.6%) by an interplay of sexual behavioural, socioeconomic, and environmental variables. The MOR of the nested model is 2.41, whereas the MOR of the final model is 1.94. This illustrates that, although some of the location-level heterogeneity is captured by the model fixed-effect covariates, almost two-thirds of the heterogeneity in HIV prevalence at sample locations still could not be explained by the covariates in the model. Overall, environmental fixed effects reduce the location-level heterogeneity more than sexual behavioural and socioeconomic fixed effects (MORs of 1.95, 2.35, and 2.30, respectively). Adjusting for country improves the model fit but does not change the importance of the different predictors and does not considerably increase the degree of explained heterogeneity (marginal $R^2$ 17.9%, conditional $R^2$ 29.3%) (see S10 Table).

The maps in Fig 3 show the interpolated random effects estimates—i.e., the unexplained heterogeneity in HIV prevalence—for the 5 models. The white areas represent locations where relatively most heterogeneity is explained by the model. The red and purple areas represent locations where relatively the least heterogeneity is explained. As expected, random effect estimates in the nested model were highest in high-prevalence areas (panel A) and decline as fixed effects are added to the models (panels B–E). Interpolated random effects estimates from the combined model (panel B) are substantially reduced. However, the geospatial heterogeneity in many areas with a high prevalence remains unexplained, for example, around Lake Victoria (1), at the major ports of Mozambique (2–4), at Plumtree (5), around Mongu (6) and the Copperbelt (7) and Nchelenge (8) districts in Zambia. In most of these locations, environmental

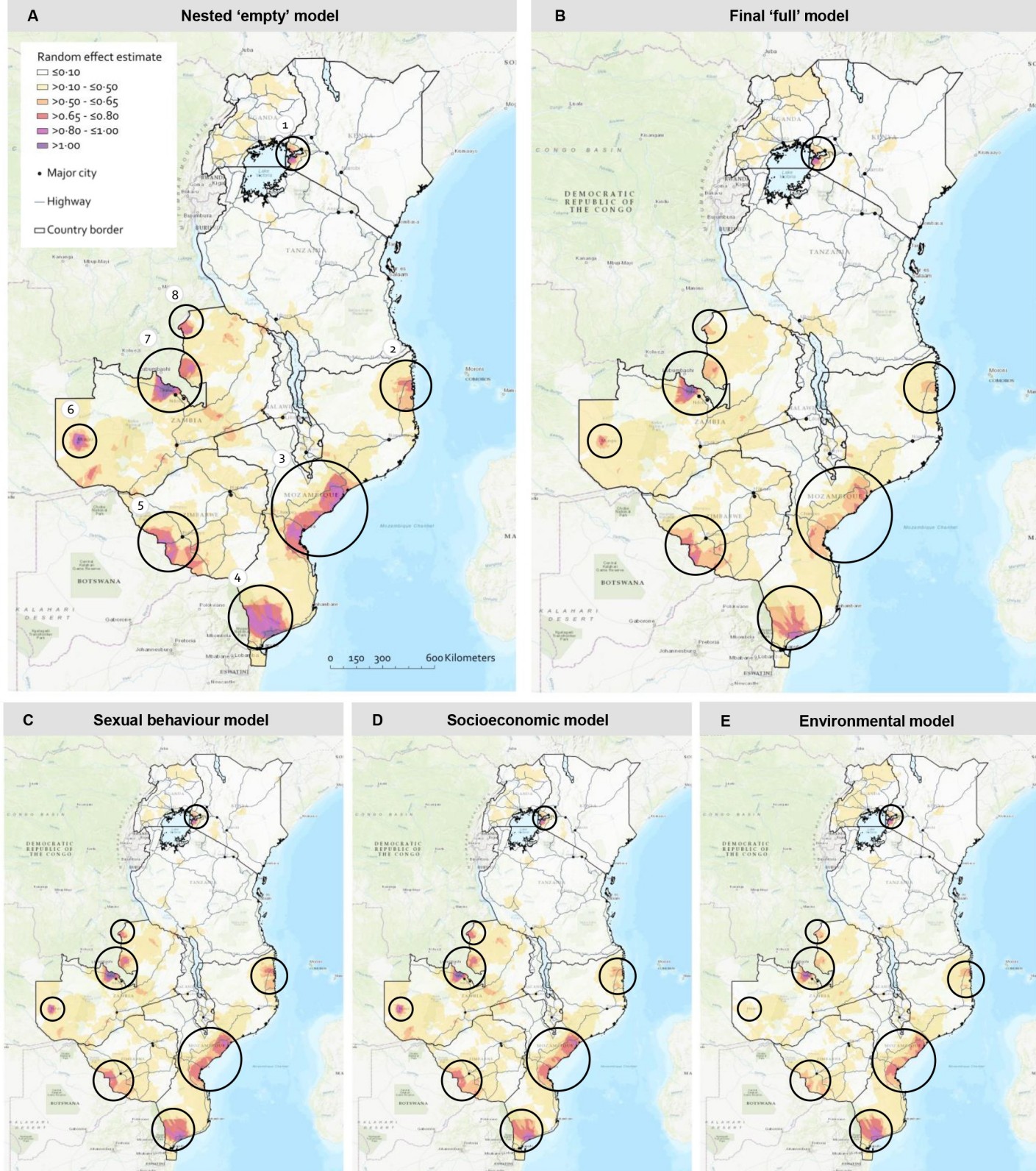

**Fig 3. Maps present the interpolated random effect estimates of the nested 'empty' logistic regression model (A), best-fitting logistic regression model (B), and the separate models including only sexual behavioural (C), socioeconomic (D), and environmental variables (E) among young adults (women 15–24 years and**

**men 15–29 years) for 7 countries in Eastern and Southern Africa.** For the nested 'empty' model (A), random effect estimates reflect HIV prevalence levels among young adults (see Fig 2B). For the other models (B–E), random effect estimates are lower at (some of the) areas with high HIV prevalence levels, indicating that the additional variables in each model to some extent explain HIV heterogeneity at these locations. Circles point out high-prevalence areas of HIV among young adults (1) around Lake Victoria, (2) around Nacala (and Pemba) port, (3) around Beira (and Quelimane) port, (4) around Maputo city and port, (5) around Plumtree border crossing, (6) around Mongu city, (7) in the Copperbelt mining area, and (8) in Nchelenge district. HIV, human immunodeficiency virus.

variables were better at explaining heterogeneity (panel E) than sexual behavioural or socio-economic variables (panels C and D, respectively).

## Discussion

Our findings showed that substantial levels of spatial heterogeneity in HIV prevalence exist among adults and young adults throughout all 7 Eastern and Southern African countries analysed in this study. Especially in young adults, micro-epidemics of relatively high prevalence alternated with areas of very low prevalence, clearly illustrating the existence of areas with high levels of transmission. Overall, 15.6% (marginal $R^2$) of the heterogeneity in HIV prevalence could be explained by an interplay of behavioural, socioeconomic, and environmental factors, including number of sex partners, STIs, GHF (a proxy for economic activity), and urbanization. Maps of interpolated random effect estimates at each sample location showed that environmental predictors were better at specifically predicting HIV prevalence at the high-prevalence areas than sexual behavioural or socioeconomic variables, yet substantial heterogeneity at other high-prevalence areas remains unexplained.

The geospatial patterns of HIV prevalence heterogeneity among adults shown in our study (Fig 2A) are very comparable to patterns shown in other recent studies in which other methods for spatial interpolation were used [4,5], confirming that our approach was suitable for creating reliable estimates. As an external validation for our geospatial patterns of HIV prevalence in young adults, we compared our estimates to the estimates from multiple small-scale surveillance sites from the ALPHA network [23]. We found that most estimates are comparable, but our estimates were lower for the Rakai (Uganda) and Manicaland (Zimbabwe) areas (S14 Fig). Both areas are characterised by well-known high levels of HIV transmission and relatively low numbers of sample locations in the utilised data with varying prevalence levels, leading to underestimation of the HIV prevalence. This indicates that our approach of kriging, in which prevalence at a specific location is estimated by the prevalence in surrounding clusters as a function of the distance between the location and surrounding clusters, may have resulted in an underestimation (i.e. smoothing) of HIV prevalence estimates, especially in high-prevalence areas.

Our findings that environmental factors were the most important determinants of geospatial heterogeneity is strengthened by the observation that all high HIV prevalence areas among young adults are in locations with known high levels of economic activity: characterised by high production and flow of goods and services [25,26]. This suggests that high-risk dynamics (involving seasonal work and commercial sex) in these areas might be important in generating this heterogeneity [27,28]. For instance, the fishing communities around Lake Victoria in Uganda (circle 1 in Fig 3) have frequently been reported as sites with high levels of transactional sex and HIV [29,30]. Furthermore, many high-prevalence areas in our analyses cluster around border crossings, major highways or major ports (e.g. circles 2 to 5 in Fig 3). Long distance truck driving and associated commercial sex have been documented as important contributors to HIV transmission [27,29]. In addition, some of the high-prevalence areas are in regions known to have high levels of migration, either work-related (seasonal) migration or other types of migration. For example, the Copperbelt mining area (circle 7 in Fig 3) or

Nchelenge district in Zambia (circle 8 in Fig 3) in Zambia, which is known for its active fishing industry and a big refugee settlement. Mining and fishing areas have long been recognised as high risk settings for HIV transmission, as domestic or foreign male workers often work long stints, separated from their families surrounded by an active sex industry [29–33].

Individual HIV prevalence in young adults was strongly associated with the reported number of lifetime sex partners and reported prevalence of STIs or STI symptoms, providing further evidence that geospatial clustering of HIV is linked to the clustering of risk behaviour, across all countries [12]. We found that the reported condom use at last sex act was associated with an increased risk of being infected with HIV. This is a well-known counterintuitive finding, and reflects the fact that condom use tends to be higher (yet sometimes insufficient) among people with riskier sexual behaviour, or reflects bias due to the fact that people who are aware of their positive HIV status are more likely to use a condom, to protect their partner [34]. This finding highlights the need to stimulate condom use, but more importantly increase access to effective HIV prevention interventions more broadly. Our finding that having education beyond the primary level seems to be strongly protective against HIV, is consistent with observations in the literature [13], and suggests that structural interventions to improve educational attainment [14] could help reduce HIV transmission in adolescent women.

Our results have important implications for the planning of prevention and treatment programs. High HIV prevalence areas in young adults are likely areas of high transmission [10], requiring prioritization of tailored prevention interventions. Our key findings that (i) high HIV prevalence areas among young adults are located at economically active or developing areas and (ii) that HIV among young adults is driven by risky sexual behaviour; indicate that preventive interventions targeted at young adults at these specific locations could strongly impact HIV transmission. Prevention programs aimed at improving exposure and uptake of effective prevention interventions for young adults [35,36]–such as pre-exposure prophylaxis (PrEP) [37], condoms [38], or voluntary male medical circumcision programmes [15,39]– should be prioritized in these areas. Also, intervention programs for young adults should aim at early diagnosis and treatment initiation of those who get infected with HIV, by creating accessible, affordable and youth-friendly HIV testing and counselling services [35,40]. Furthermore, governments typically have prior knowledge about future economic developments. Improving the resilience of affected populations against the associated health risks, might be considered an integral part of such development. Moreover, as the population in these areas is likely to be highly mobile [41,42], effective prevention for young adults in high-prevalence areas may not only affect the local HIV epidemics, but also the wider epidemic. Ultimately, our results demonstrate that the decade-old mantra of "know your epidemic, know your response" [43] is still highly relevant for SSA. Our study found important common denominators that are associated with increased HIV risk in areas with high levels of transmission, but it will be essential for policy makers to specifically evaluate the behavioural and socioeconomic context, and the interventions already in place at a specific high-risk setting to tailor interventions appropriately.

To our knowledge, we are the first to utilise geolocated HIV prevalence data from young adults specifically, to explore geospatial heterogeneity in the HIV epidemics of Eastern and Southern Africa, and identify potential areas with high levels of transmission. Previously, Cuadros et al proposed a co-kriging approach to estimate subnational HIV prevalence estimates, incorporating HIV prevalence and environmental determinants [4], yet they did not stratify by age. In addition, Palk and Blower recently showed that places of high HIV prevalence in adults (aged 15–49 years) in Malawi are associated with reported higher rates of high-risk sex, defined as the number of lifetime partners [12]. Our results show that heterogeneity in HIV is associated with a range of sexual behavioural, socioeconomic and environmental variables,

and that these are highly affected by age. Therefore, a standardised approach to map heterogeneity of an HIV epidemic should take age stratifications into account, and future studies on geospatial heterogeneity and its drivers should include behavioural, socioeconomic and environmental determinants.

Our results have limitations. First, DHSs produce cross-sectional data intended to provide a measure of HIV prevalence and behaviour among the general population; and thus, high-risk subpopulations such as female sex workers, men who have sex with men, and mobile populations such as truck drivers and seasonal workers, are thought to be underrepresented in these surveys. Therefore, we cannot make definitive conclusions on economic activity and key populations driving HIV transmission in at these locations. Nevertheless, our analyses showed that high HIV prevalence areas in DHS data, especially among young adults, were almost invariably located near economically active areas suggesting that these key-population dynamics are still visible through general population-based surveys. Extending DHSs with other data sources that allow for mapping of key populations or performing incidence essays on HIV samples will allow for more accurate identification of HIV transmission areas and can further enhance our understanding of the epidemic. Second, DHS sampling locations are randomly selected, based on the underlying population density within a country. Consequently, very few locations were sampled from areas with large nature and wildlife conservation reserves, such as in Northern Kenya, Central Tanzania, and Northern and South-western Mozambique, and our interpolated prevalence estimates for such areas should be interpreted with caution. Designing alternative sampling techniques that over-sample areas with low densities could increase reliability of interpolated survey results. Third, we used survey data from a relatively wide range of years; 2008–2009 to 2015–2016. This time period in the study coincides with major initiatives to curb the pandemic, in particular the scale-up of antiretroviral treatment, as well as voluntary male circumcision campaigns and other HIV prevention interventions. Although these initiatives were potentially disproportionally targeted at high transmission areas, we expect that the disproportionate impact would not be so extreme that it completely alter the locations of high transmission areas. Furthermore, although the scale-up of these interventions may have reduced HIV incidence, the associations between HIV and the hypothesised main drivers of HIV transmission–the (sexual) behavioural, socioeconomic and environmental factors explored in this study–likely did not change substantially by these interventions. Fourth, explorative statistical analyses always run the risk of identifying patterns of random noise [44]. However, we believe this risk to be extremely low in our study, due to the very large sample size ($n = 53,234$), the rigid preselection of variables based on substantive knowledge, and the directions and magnitudes of association between the included covariates and HIV found in our study are in line with of findings from previous studies [2–5]. In addition, we performed nonrandom cross-validation by testing the final fitted model for each country separately. Reassuringly we found that, despite the differences in underlying epidemic and scale-up of interventions across countries, the conditional $R^2$ of the model was strikingly similar for six out of the eight countries in our analysis, ranging between 23.5% and 36.0% (compared to 29.6% in the main analysis). Only the conditional $R^2$ for Kenya (45.0%) and Zimbabwe (17.6%) deviated a little bit more from the combined model, yet not to an alarming extend (S11 Table).

In conclusion, we our findings show that consistent clustering of HIV prevalence exists among young adults in seven high burden countries in Eastern and Southern Africa, with clearly identifiable high-prevalence areas. This heterogeneity is driven by an interplay of behavioural, socioeconomic and environmental factors, and the locations of high-prevalence areas suggest that key population dynamics, especially related to seasonal and economic migration and associated sex work, play a major role. In further reducing HIV transmission in Eastern and Southern Africa, areas of high HIV prevalence in young adults could be priority areas for

tailored HIV prevention interventions in line with SDG3 and UNAIDS targets to end the HIV pandemic by 2030.

## Supporting information

**S1 Checklist. The RECORD statement—Checklist of items, extended from the STROBE statement, that should be reported in observational studies using routinely collected health data.**
(DOCX)

**S1 Equations. Mathematical equations of our kriging model and MOR calculations.**
(DOCX)

**S1 Fig. Overview of the study area in SSA (top right panel) and the DHS and AIS sample locations (blue dots) for the 7 countries included in this study: Kenya ($n$ = 394), Malawi ($n$ = 847)[1], Mozambique ($n$ = 270), Tanzania ($n$ = 570)[1], Uganda ($n$ = 470)[1], Zambia ($n$ = 719), and Zimbabwe ($n$ = 400).** [1]AIS.
(PDF)

**S2 Fig. Map presents whether the sample location is classified as urban or rural.**
(PDF)

**S3 Fig. Map presents the population density for the study area.**
(PDF)

**S4 Fig. Map presents the proximity from each sample location to the nearest highway.**
(PDF)

**S5 Fig. Map presents the proximity from each sample location to the nearest major city (more than 250,000 inhabitants).**
(PDF)

**S6 Fig. Map presents the proximity from each sample location to the nearest border crossing or major port.**
(PDF)

**S7 Fig. Map presents the EVI for the study area.**
(PDF)

**S8 Fig. Map presents the GHF for the study area.**
(PDF)

**S9 Fig. Maps present the predicted HIV prevalence in women (15–49 years) (A) and men (15–54 years) (B) for 7 countries in Eastern and Southern Africa.** The maps of HIV prevalence among adults and young adults are shown in Fig 2. Continuous surface maps were created by kriging HIV prevalence data obtained from (https://dhsprogram.com/).
(PDF)

**S10 Fig.** Maps and scatterplots illustrating the difference in HIV prevalence (per 5 km$^2$ grid cell) between women and men (A and C, respectively) and between adults and young adults (B and D, respectively), for 7 countries of Eastern and Southern Africa.
(PDF)

**S11 Fig.** Density plots illustrating the overall sample location-level distributions of HIV prevalence among adults (A), young adults (B), women (C), and men (D) for each country included

in this study.
(PDF)

**S12 Fig. Density plots illustrating the overall logit-transformed sample location-level distributions of HIV prevalence among adults (A), young adults (B), women (C), and men (D) for each country included in this study, as used for semivariogram modelling and ordinary kriging.** The logit-transformed HIV prevalence of −6 (on the x-axis) represents a prevalence of 0%, −5 of 1%, −4 of 2%, −3 of 5%, −2 of 12%, −1 of 27%, 0 of 50%, and 1 of 73.
(PDF)

**S13 Fig.** Plots illustrating the observed versus the predicted sample location-level HIV prevalence (A) and the observed versus the predicted number of HIV cases (B) among young adults (women 15–24 years and men 15–29 years) for the combined 'full' best-fitting multiple multilevel regression model per DHS sample location for 7 countries of Eastern and Southern Africa (also see S9 Table).
(PDF)

**S14 Fig. Map of HIV prevalence estimates for young adults, as interpolated in this study, and HIV prevalence estimates for young adults as reported from 7 population-based cohorts at small-scale geographical sites (ALPHA network) within the area covered by this study.**
(PDF)

**S1 Table. Overview of all variables included in the study.**
(DOCX)

**S2 Table. Spatial autocorrelation of HIV prevalence at the sample location level, estimated by Moran's *I* index.**
(DOCX)

**S3 Table. Bivariate logistic regression models of HIV status and behavioural, socioeconomic, and environmental variables in young adults (women 15–24 years and men 15–29 years of age) in 7 countries of Eastern and Southern Africa, adjusted for age and sex.** Data obtained through (https://dhsprogram.com/).
(DOCX)

**S4 Table. Multiple multilevel nested 'empty' logistic regression model of HIV status in young adults (women 15–24 years and men 15–29 years of age) for 7 countries of Eastern and Southern Africa, adjusted for age and sex.** Data obtained through (https://dhsprogram.com/).
(DOCX)

**S5 Table. Multiple multilevel logistic regression model of HIV status and behavioural variables in young adults (women 15–24 years and men 15–29 years of age) for 7 countries of Eastern and Southern Africa, adjusted for age and sex.** Data obtained through (https://dhsprogram.com/).
(DOCX)

**S6 Table. Multiple multilevel logistic regression model of HIV status and socioeconomic variables in young adults (women 15–24 years and men 15–29 years of age) for 7 countries of Eastern and Southern Africa, adjusted for age and sex.** Data obtained through (https://dhsprogram.com/).
(DOCX)

**S7 Table. Multiple multilevel logistic regression model of HIV status and environmental variables in young adults (women 15–24 years and men 15–29 years of age) for 7 countries of Eastern and Southern Africa, adjusted for age and sex.** Data obtained through (https://dhsprogram.com/).
(DOCX)

**S8 Table. Combined 'full' multiple multilevel logistic regression model of HIV status and behavioural, socioeconomic, and environmental variables in young adults (women 15–24 years and men 15–29 years of age) for 7 countries of Eastern and Southern Africa, adjusted for age and sex.** Data obtained through (https://dhsprogram.com/).
(DOCX)

**S9 Table. Combined 'full' multiple multilevel model as modified Poisson regression (with robust variance) of HIV status and behavioural, socioeconomic, and environmental variables in young adults (women 15–24 years and men 15–29 years of age) for 7 countries of Eastern and Southern Africa, adjusted for age and sex.** Data obtained through (https://dhsprogram.com/).
(DOCX)

**S10 Table. Combined 'full' multiple multilevel logistic regression model of HIV status and behavioural, socioeconomic, and environmental variables in young adults (women 15–24 years and men 15–29 years of age) for 7 countries of Eastern and Southern Africa, adjusted for age, sex, and country.** Data obtained through (https://dhsprogram.com/).
(DOCX)

**S11 Table. Overview of the heterogeneity ($R^2$) explained by the full final logistic regression model (see S9 Table), for each country separately.**
(DOCX)

## Acknowledgments

The authors thankfully acknowledge Daan Nieboer, statistician at the Erasmus Medical Center, for the helpful discussions regarding statistical methods.

## Author Contributions

**Conceptualization:** Caroline A. Bulstra, Jan A. C. Hontelez, Richard Steen, Nico J. D. Nagelkerke, Till Bärnighausen, Sake J. de Vlas.

**Data curation:** Caroline A. Bulstra.

**Formal analysis:** Caroline A. Bulstra, Federica Giardina.

**Funding acquisition:** Jan A. C. Hontelez, Sake J. de Vlas.

**Investigation:** Caroline A. Bulstra, Jan A. C. Hontelez, Federica Giardina, Richard Steen, Sake J. de Vlas.

**Methodology:** Caroline A. Bulstra, Jan A. C. Hontelez, Federica Giardina, Sake J. de Vlas.

**Supervision:** Jan A. C. Hontelez, Nico J. D. Nagelkerke, Till Bärnighausen, Sake J. de Vlas.

**Validation:** Caroline A. Bulstra, Federica Giardina, Sake J. de Vlas.

**Visualization:** Caroline A. Bulstra, Sake J. de Vlas.

**Writing – original draft:** Caroline A. Bulstra, Jan A. C. Hontelez.

**Writing – review & editing:** Caroline A. Bulstra, Jan A. C. Hontelez, Federica Giardina, Richard Steen, Nico J. D. Nagelkerke, Till Bärnighausen, Sake J. de Vlas.

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
