## [Decision Letter · Decision Letter 0]

13 Nov 2019

Dear Dr. Hontelez,

Thank you very much for submitting your manuscript "Mapping and characterising HIV transmission hotspots in sub-Saharan Africa: a geospatial analysis of national survey data" (PMEDICINE-D-19-03490) for consideration at PLOS Medicine. 

Your paper was discussed with an academic editor with relevant expertise and sent to independent reviewers, including a statistical reviewer. The reviews are appended at the bottom of this email and any accompanying reviewer attachments can be seen via the link below:

[LINK]

In light of these reviews, we will not be able to accept the manuscript for publication in the journal in its current form, but we would like to invite you to submit a revised version that fully addresses the reviewers' and editors' comments. You will appreciate that we cannot make a decision about publication until we have seen the revised manuscript and your response, and we expect to seek re-review by one or more of the reviewers. 

We hope to receive your revised manuscript by Dec 04 2019 11:59PM. Please email us (plosmedicine@plos.org) if you have any questions or concerns.

Please let me know if you have any questions. Otherwise, we look forward to receiving your revised manuscript in due course. 

Sincerely,

Richard Turner, PhD

rturner@plos.org

Our academic editor commented that the term "hotspot" is not favoured by people with HIV, and we ask you to amend this. 

Also, it was suggested that "the framing could be that by addressing areas of high incidence we are addressing inequities rather than focusing on areas perceived to be driving the epidemic".

In your abstract, please specify the range of years that the source datasets refer to. Also, please add summary demographic details for study participants. 

Please add a new final sentence to the "methods and findings" subsection of your abstract to summarize the study's main limitations. 

At line 40, please start the "Conclusions" subsection with "In this study, we found that ..." or similar. 

After the abstract, we ask you to add a new and accessible "author summary" section in non-identical prose. You may find it helpful to consult one or two recent research papers published in PLOS Medicine to get a sense of the preferred style. 

In your methods section, please briefly mention the situation with ethics approval (e.g., not required for this analysis). 

Early in the methods section of your main text, please state whether the study had a protocol or prespecified analysis plan, and if so attach the relevant document(s) as a supplementary file (referred to in the methods). Please highlight analyses that were not prespecified. 

Please add a completed checklist for the most appropriate reporting guideline, which may be RECORD, as a supplementary file (again referred to in your methods section. In the checklist, individual items should be referred to by section (e.g., "Methods") and paragraph number rather than by line or page numbers, as the latter generally change in the event of publication. 

When discussing your conclusions, e.g. in the first paragraph of your Discussion section, please ensure that findings are referred to consistently in the past tense (e.g., "alternated" at line 268) with conclusions in the present tense (e.g., "Our findings show ..." at line 266). 

Please avoid claims of primacy (e.g., at line 317), and where used add "to our knowledge" or similar. 

Please remove "(80-)" from reference 17. 

Comments from the reviewers:

*** Reviewer #1: 

I confine my remarks to statistical aspects of this paper. I have a couple of fairly major issues to resolve before I can recommend publication, but I think there's a good paper here.

My first question is what was done to avoid finding patterns in pure noise? How can we be sure that these patterns are not just things found by the power of the method?

The second big thing is the model formation. The authors did bivariate screening followed by stepwise methods. Both of these are problematic. All of the output from such methods is wrong: Parameter estimates are biased away fro 0, p values are too small, standard errors are too small (see Harrell **Regression Modeling Strategies* for lots of details). It would be better to use substantive knowledge to build a model. If the authors insist on not using their knowledge and want an automatic method, the LASSO is probably the best of the readily available methods.

More minor points

The authors should list all the variables and how they were measured. Not just the ones that wound up in the model, but all the ones that were available

Line 139 "Multivariate" should be "multiple"

Line 144 Age and sex should be fixed effects

Line 153 and other places R^2 is mistyped as R2

Line 158 I don't understan how random effects were translated into MOR. First, random effects are usually nuisance variables that you are not particularly interested in. Second, the standard errors for random effects are very poorly estimated (so much that some programs don't even print them) so I think the authors mean fixed effects. Third, I don't know how the translation was done.

Figure 1 How were the outliers dealt with?

Peter Flom

*** Reviewer #2: 

This work attempts to map 'hotspots' for young adults (proxy for higher transmission hotspots for targeting) in seven countries in Eastern and Southern Africa, specifically identify significant clustering and associated determinants potentially explaining the underlying heterogeneity observed. The work I believe makes a sufficient contribution/step forward to the existing body of literature in this area, is well written and should be useful for policy makers in these countries. I have some comments and suggestions below that need to be addressed before the paper can be accepted.

General: Identifying the hotspots is first natural step as currently presented in the paper. However the current detailed description of explained versus unexplained heterogeneity explained for the larger covariate groupings in the main results text is not in a policy maker friendly format and needs to be revised to maximise the readability/usability of these results. It would be more useful to present key findings highlighting which risk factors or covariates are the most attributable both within and across countries to assist with geographic/locally tailored intervention packages.

General: It is also important present the spatial hotspot/interpolation surface by gender in addition to overall (e.g. SDG 5: Gender equality) to confirm if the same hotspots are identified for both males and females. It may also be worth considering shared component analysis of the gender specific prevalence surfaces to see to which degree they correlate at finer geographic scale and how this varies within and across the countries.

General: Apologies if missed it but there is little or no contextualisation of the results with regards to SDG targets for HIV and how these results are relevant to assessing progress towards these targets and assisting with retailoring of intervention packages across the individual countries in this analysis.

Major comments:

1) The analysis utilised data from DHS/AIS surveys, however there is also a wealth of small geographic scale data in the demographic surveillance sites (DSS) in the region and ongoing HIV work in these sites e.g. ALPHA network. I was wondering if the authors considered using prevalence estimates from these DSS as an external validation source for the applied kriging exercise? i.e. does the model predict well at unsampled locations that potentially overlap with DSS.

2) Did the authors consider a Bayesian spatial approach/implementation in R-INLA as opposed to ordinary kriging? A spatial Bayesian multivariable formulation including the covariates might then be able to produce uncertaintiy around predictions at unsampled locations and more efficiently adjust for spatial correlation in nearby sampled clusters as per DHS design. I suspect that a Bayesian approach in INLA (which would be computationally tractable) would perform better than the approaches currently employed in the paper.

3) For the risk factor analysis/modelling I can understand not using the original sampling weights from the DHS/AIS. However when estimating prevalence and generating a smoothed prevalence surface I would suspect that these weights be more important and allow uncertainty intervals (e.g. Figure 1 etc). Did the authors perform a sensitivity analysis to confirm that the surface without weights would be fairly similar to a surface utilising the survey weights? The weights would also be useful if you were to add additional visualisations projecting absolute counts of HIV positive young adults by high versus lower risk clusters. Prevalence without weighted correction may mask underlying population size differences within and across countries.

4) The environmental covariates if I understand it are homogeneous for all individuals within a DHS survey "cluster" or community while those measured at individual level varying at that level. I wonder if this may impact the variable selection approach, especially if income or SES at household level is linked up with GHF (economic activity component) at cluster level.

5) Population attributability - it may be worth considering a Poisson model with robust variance to estimate relative risks and the leverage prevalence of exposure to various risk factors to estimate population attributable fractions for each in addition to the effect sizes currently presented. This may reveal some additional subtle differences across the countries and within. Additional a decomposition type approach (e.g. Shapley) using a generalised linear framework could also be used to estimate the relative importance of the various covariates.

7) Results narrative - i think the explanation of the directionality of association for the environmental covariate e.g. EVI and GHF needs to be improved (e.g. lines 216-217).

8) If I understand Figure 3 correctly the least heterogeneity is explained in the areas with the highest prevalence among young adults (Figure 2)? This is important and has implication for the conclusions/utility of the results.

Minor

1)Risk factor analysis: S2 Table - the coefficient for gender looks incorrect i.e. 0.00? Especially given the highly significant coefficient in S3 Table.

*** Reviewer #3: 

This well-organized study uses geospatial methods to identify areas of high HIV prevalence among young adults within 7 countries in Eastern and sub-Saharan Africa. Additionally, the authors present a regression model that relates epidemiological predictors to HIV prevalence. While the approach is sound, the significance of is more of an incremental contribution rather than a major advance from published geospatial modeling studies of HIV prevalence, such as the works by Cuadros et al and Dwyer-Lindgren et al that are cited in the introduction of the paper. The focus on HIV prevalence among young adults is interesting and could be developed further. 

Major critiques:

1. The paper focuses on HIV prevalence among young adults, with the claim that young adults represent more recent infection and that high prevalence locations can be prioritized for prevention interventions. This is an interesting focus and could be developed further in the paper. Are the locations of high HIV prevalence in young adults the same as the locations among the general adult population? How do areas of high prevalence among young adults compare with variation in the underlying population age structure? More can be said in the discussion about efforts to prevent HIV infection specifically in this age group.

2. Line 171-172 and Figure 1. How were the national-level prevalence estimates generated? Are these the weighted means of the clusters values or do they use the estimated HIV prevalence rasters? How do they compare with the published survey reports? It would enhance the results to provide a confidence interval around the estimate.

3. The nine-year timespan over which the survey data were collected means that the resulting maps are a composite of various survey years. The years represented in the study coincided with a major scale-up of ART and HIV prevention interventions, so the difference in timing could be important. The difference in survey years should be included in the limitations. 

4. The claim in the discussion lines 278-279 that all high prevalence areas were areas with known high levels of economic activity needs to be tested more rigorously. How is the level of economic activity for an area defined in this analysis? Was the level of economic activity assessed over the entire geographic area? Were areas also found that had high economic activity and low HIV prevalence?

Minor critiques

1. Please make it clear which years are represented for the covariates that were not directly extracted from the DHS or AIS surveys (eg WorldPop).

2. Include a justification for why a larger age group was defined as young adults for males than for females.

3. Line 124 statistical analysis. Please provide the interpolation equations in the supplementary material. 

4. How does the amount of variation explained by the models in this study compare to R2 in other published studies?

5. Line 300. I do not agree that the findings of this study support the need to increase condom use specifically. Rather, persistently high HIV incidence supports the need to increase access to effective HIV prevention interventions more broadly.

6. Line 317, "We are the first…". Please modify this statement in light of the other published HIV prevalence maps cited in the paper. While this paper's focus on young adults takes a new lens to these data, they are fully utilized in the other studies as well.

***

[LINK]

---

## [Decision Letter · Decision Letter 1]

13 Jan 2020

Dear Dr. Hontelez,

Thank you very much for re-submitting your manuscript "Mapping and characterising areas with high levels of HIV transmission in sub-Saharan Africa: a geospatial analysis of national survey data" (PMEDICINE-D-19-03490R1) for consideration at PLOS Medicine.

I have discussed the paper with editorial colleagues and our academic editor, and it was also seen again by three reviewers. I am pleased to tell you that, provided the remaining editorial and production issues are dealt with, we expect to be able to accept the paper for publication in the journal.

[LINK]

Please let me know if you have any questions. Otherwise, we look forward to receiving your revised manuscript shortly. 

Kind regards,

Richard Turner, PhD

rturner@plos.org

Requests from Editors:

In the abstract, please present numbers in the format of "113,000 adults" at line 32, for example.

Around line 33, again in the abstract, we suggest adding a sentence to quote the range of mean levels of HIV prevalence by country, for adults and young adults, to contrast the local numbers quoted subsequently.

Also at lines 32 and 247, please make that "... of which about 53,000 were young adults ..." or similar.

At line 35, we suggest adapting the current wording to " ... among young adults as high as 11% or 15%"

At line 59, the wording "Heterogeneity could be explained for 15.6% ..." suggests "15.6 of participants", and we ask you to reword this text if you are referring to "15.6% of heterogeneity" (and similarly at line 367). 

In the abstract, you may wish to quote additional findings from column 3 of table 2 in a new sentence, say, to help readers appreciate the context of the quoted value of "15.6%". 

At line 42, please make that "main study limitations". 

Should "eSwatini" be substituted at line 90?

At line 239, please make that "followed RECORD guidelines". 

At line 316, should that be "7.2% (marginal R2) or 26.3% (conditional R2) of the HIV heterogeneity ..."?

At line 320, please add a comma ("... sexual, behavioural ..."). 

At line 363, please make that "substantial levels ... exist among ...". 

At line 368, please add a few words to explain what "global human footprint" is.

At line 453, please amend the text to "... DHS surveys produce cross-sectional data intended to provide a measure of HIV prevalence and behavior ..." or similar. 

Please add fuller access details to reference 11, as available. 

Please remove "[Internet]" from reference 16 and any other relevant references, and add a cited date. 

Please convert the RECORD checklist to a separate supplementary file, named "S1_Checklist". 

Comments from Reviewers:

*** Reviewer #1:

The authors have addressed my concerns and I now recommend publication

Peter Flom

*** Reviewer #2:

[supportive report received]

*** Reviewer #3: 

My requests in the initial review were adequately addressed by the authors.

***

[LINK]

---

## [Editor Report · Decision Letter 2]

5 Feb 2020

Dear Dr. Hontelez, 

On behalf of my colleagues and the academic editor, Dr. Ruanne Barnabas, I am delighted to inform you that your manuscript entitled "Mapping and characterising areas with high levels of HIV transmission in sub-Saharan Africa: a geospatial analysis of national survey data" (PMEDICINE-D-19-03490R2) has been accepted for publication in PLOS Medicine. 

PRODUCTION PROCESS

PRESS

PROFILE INFORMATION

Thank you again for submitting the manuscript to PLOS Medicine. We look forward to publishing it. 

Best wishes, 

Richard Turner, PhD

Senior Editor 

PLOS Medicine

plosmedicine.org